# Evaluation and Source Identification of Heavy Metal Pollution in Black Soils, Central-Eastern Changchun, China

Yaoyao Sun , Yuyan Zhao, Libo Hao, Xinyun Zhao *, Jilong Lu , Qiaoqiao Wei *, Yanxiang Shi and Chengyou Ma

Department of Geochemistry, Jilin University, Changchun 130021, China; yaoyaos19@mails.jlu.edu.cn (Y.S.); zhaoyuyuan@jlu.edu.cn (Y.Z.); haolb@jlu.edu.cn (L.H.); lujl@jlu.edu.cn (J.L.); shiyx@jlu.edu.cn (Y.S.); chengyou@jlu.edu.cn (C.M.)
* Correspondence: zhaoxy15@jlu.edu.cn (X.Z.); weiqiaoqiao888@sina.com (Q.W.)

**Abstract:** Black soils are vital agricultural resources, and assessing heavy metal contamination in black soils is of great significance to the sustainable development of agriculture and the environment. In this study, 1246 surface soil samples were collected from the central-eastern part of Changchun, where phaeozems and chernozems are widely distributed, and the As, Hg, Cr, Cd, and Pb concentrations were determined to investigate the pollution status in the black soils by the geoaccumulation index ($I_{geo}$). To eliminate the influence of background variation and improve the calculation accuracy of the $I_{geo}$ values, the local background values of these five elements were estimated after classifying the samples into three clusters with the $k$-means clustering method. The $I_{geo}$ calculated with the local background values not only identified the pollution that is difficult to recognize in the low-background areas but also eliminated the easily misidentified pollution in the high-background areas. The $I_{geo}$ results show that the black soils are mainly contaminated with Hg, followed by Cd and Pb, and are almost free from the pollution of As and Cr. The further the sampling sites are from urban areas, the milder the soil pollution is. A positive matrix factorization (PMF) analysis shows that industrial activities and coal burning contributed the most to the heavy metal pollution in the black soils, followed by agricultural activities, which should be paid more attention to.

**Keywords:** heavy metal pollution; black soils; geoaccumulation index; background variation; $k$-means clustering method

## 1. Introduction

Black soils, known as mollisols in the United States system of soil taxonomy, have a thick and dark-colored surface horizon, resulting from humus enrichment from litterfall decomposition [1]. Black soils are fertile and suitable for planting, containing large quantities of nutrients and organic matter [2,3]. There are only four major black soil regions worldwide, including the Ukrainian plain, the Mississippi River basin of North America, northeastern China, and the Pampas of South America [4]. With the rapid development of industry and agriculture, however, black soils may suffer from different degrees of heavy metal pollution, which poses a certain degree of threat to the sustainable development of black soils [5–7].

Heavy metals generally refer to metals with a density greater than 4.5 g/cm$^3$, and the definition varies from discipline to discipline due to different research emphases. In environmental research, the term mainly refers to metals that have significant biological toxicity, such as arsenic (As), mercury (Hg), chromium (Cr), cadmium (Cd), and lead (Pb). Arsenic is considered a metal mainly due to its metal-like nature [8]. The accumulation of these heavy metals in soils significantly impacts human health and the ecosystem [9,10]. For example, rice crops cultivated in Cd-contaminated soils were found to uptake Cd, and consuming the rice can cause Cd poisoning [11]. This fact leads to the necessity of investigating the accumulation of heavy metals in black soils.

The heavy metal pollution in the black soils has increasingly become a focus [12]. The background value is necessary for soil pollution assessment [13–15]. There are, however, many factors that may have an impact on the calculation of the element background in soils, and one of the most considerable factors may be that of the lithology background of the soil parent materials, which can cause the element background to vary from place to place [16,17]. Obviously, the element background should not always be considered a uniform value in lithologically complex regions [18–20]; otherwise, it can mislead the utilization and conservation of soil resources.

To calculate the local background values for an element in soils, the lithology background of the soil parent materials should be separated first. However, it is impossible to separate the lithology background simply by dividing a region into different subareas according to the type of soil parent materials, as a parent material may have various lithologies. Since soils have a significant compositional inheritance from their bedrocks [20–22], the chemical composition of soils can be used to identify the lithology of the parent materials. Specifically, the process of distinguishing the lithology background employs an appropriate clustering method to classify the data of the elements that can reflect lithologies into different groups according to specific criteria.

This study determined the concentrations of As, Hg, Cr, Cd, and Pb in the black soil samples collected from the central-eastern part of Changchun to evaluate the heavy metal pollution. The specific objectives were to (1) remove the lithology influence by the *k*-means clustering method; (2) calculate the local background values of the As, Hg, Cr, Cd, and Pb in the black soils; (3) investigate the pollution status of these five heavy metals with the application of the geoaccumulation index ($I_{geo}$), and (4) quantify the source of these five heavy metals by the positive matrix factorization (PMF) model. The *k*-means clustering algorithm has a wide range of applications in data clustering, pattern recognition, financial risk control, and intelligent marketing. To our knowledge, it has yet to be applied to estimate the local background metal concentrations in soils. All this information can provide a reference for land conservation and sustainable management in the study area.

## 2. Materials and Methods

### 2.1. Study Area and Samples

The study area is located in the central-eastern part of Changchun, Jilin Province, northeastern China, where black soils are widely distributed (Figure 1). The black soils in the study area mainly include phaeozem, chernozem, and alluvial soils. These soils develop mainly from transported (i.e., non-residual) materials, including alluvium, diluvium, and lacustrine deposits. Their parent materials originated mainly from different types of bedrock, including granite, andesite, basalt, and coarse-grained clastic rocks [23]. Under a humid continental climate, the study area has a mean temperature of 4.6 °C and total annual precipitation of 600 to 700 mm. The primary industry of Changchun is planting, and the secondary is industrial production. Land-use patterns in the study area include agricultural, industrial, and residential uses [2].

A total of 1246 surface soil samples were collected from the study area, including 784 phaeozem samples, 348 alluvial soil samples, and 114 chernozem samples (Figure 1). The sampling density was about 1 sample per 1 km², and four neighboring small samples were consolidated to form a single large sample. The sampling depth was about 0 to 20 cm. Plant litter on the surface was scraped off during sampling. Each sample weighed more than 1 kg, and the weight reached 500 g after passing through a 0.85 mm nylon mesh.

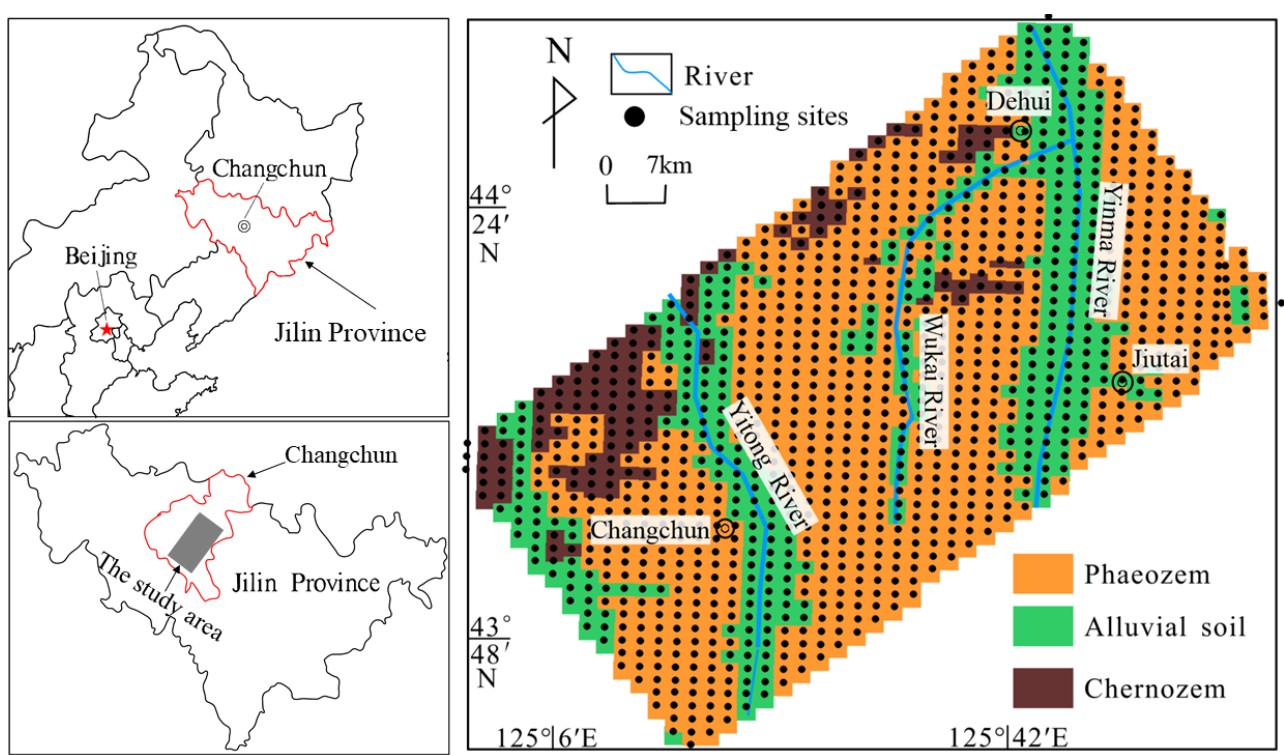

**Figure 1.** The location of the study area and sampling sites.

*2.2. Sample Pre-Treatment and Analysis*

Fifteen elements were analyzed in this study, including $SiO_2$, $Fe_2O_3$, MgO, CaO, As, Hg, Cr, Cd, Pb, Zr, Y, Sr, Sn, N, and P, and the analytical methods are shown in Table 1. Chromium, Pb, P, Zr, Sr, Y, $SiO_2$, $Fe_2O_3$, MgO, and CaO were determined by X-ray fluorescence (XRF, Tianrui EDX6000B, Beijing, China) spectrometry, and the pressed powder pellet technique was used for sample preparation. Arsenic and Hg were analyzed by atomic fluorescence spectrometry (AFS, Haiguang AFS-230E, Beijing, China). Put 0.2 g sample in a 50 mL colorimetric tube and add 10 mL aqua regia ($HNO_3$: HCl =1:3) to dissolve it in a boiling water bath (100 °C) for 2 h. Then, add 3 mL hydrochloric acid (HCl $\rho = 1.19$ g/mL), 5 mL thiourea (5% $CH_4N_2S$), and 5 mL ascorbic acid (5% $C_6H_8O_6$) and dilute with water to the scale. The sample solution was reduced by 1% $KBH_4$ and 0.2% KOH before analysis. Cadmium and Sn were determined by graphite furnace atomic absorption spectrometry (GF-AAS, Puxi A3-AFG, Beijing, China) and inductively coupled plasma atomic emission spectrometry (ICP-AES, Shimadzu ICPE-9800, Kyoto, Japan), respectively. Weigh 0.5 g soil sample in a 25 mL polytetrafluoroethylene (PTFE) crucible, add 10 mL HCl ($\rho = 1.19$ g/mL), and heat on an electric plate (<450 °C) for 2 h; then, add 15 mL nitric acid ($HNO_3$, $\rho = 1.42$ g/mL), continue heating until 5 mL of the solution remains, add another 5 mL hydrofluoric acid (HF, $\rho = 1.49$ g/mL) and 5 mL perchloric acid ($HClO_4$, $\rho = 1.68$ g/mL), and heat until the white smoke disappears. When cooled, dissolve in 1ml $HNO_3$ ($HNO_3$: $H_2O$ = 1:5) and transfer to a 50 mL volumetric bottle. Nitrogen (N) was determined by volumetry (VOL). The indicator's color change indicated the endpoint of the titration, and then the content of N in the soil samples was calculated. During the analysis, 12 National Soil Class I reference materials (GBW) (GSS-1, 2, 3, and 8–16) were inserted into every 500 samples and analyzed with the samples. The logarithmic difference and relative standard deviation between the mean and standard values were calculated to monitor the accuracy and precision of the analytical method, respectively. The analysis precision was generally better than 5% and 10% for major and trace elements, respectively.

**Table 1.** The analytical methods and detection limits.

| Elements | Analytical Methods | Detection Limits |
|---|---|---|
| Cr | XRF | 1 |
| Pb | | 1 |
| P | | 10 |
| $SiO_2$ | | 0.1% |
| $Fe_2O_3$ | | 0.04% |
| MgO | | 0.04% |
| CaO | | 0.02% |
| Zr | | 1 |
| Sr | | 1 |
| Y | | 1 |
| As | AFS | 0.9 |
| Hg | | 0.003 |
| Cd | GF-AAS | 0.02 |
| Sn | ICP-AES | 1 |
| N | VOL | 18 |

XRF—X-ray fluorescence; AFS—atomic fluorescence spectrometry; GF-AAS—graphite furnace atomic absorption spectrometry; ICP-AES—inductively coupled plasma atomic emission spectrometry; VOL—volumetry. The units of $SiO_2$, $Fe_2O_3$, MgO, and CaO are percentage, and those of other elements are mg/L.

### 2.3. Spatial Interpolation Method

In this study, the ordinary kriging interpolation analysis, a commonly used geostatistical spatial interpolation technique, was adopted to determine spatial distribution patterns of those heavy metals. It uses semi-variance to measure the degree of spatial dependence between known points [24–26], which can be calculated as follows.

$$\gamma(h) = \frac{1}{2n}\sum_{i=1}^{n}[z(x_i) - z(x_i + h)]^2 \tag{1}$$

where $\gamma(h)$ is the semi-variance at a distance interval $h$, which is the separation distance between known points; $n$ is the total number of sample value pairs separated by $h$; and $z$ is the measured sample values at location $x_i$ or $x_i + h$ [27]. The spatial distribution of heavy metals was mapped using ArcGIS 10.8, and the percentile was used to set the color level.

### 2.4. K-Means Clustering Method

In this study, the widely used $k$-means clustering algorithm [28], an important unsupervised machine learning method, was employed to eliminate the lithology influence by classifying the black soil samples in the study area. This algorithm partition objects into clusters that share similarities and are dissimilar to the objects belonging to another group, and the clustering process is as follows. Initially, $k$ clustering centers are randomly selected as centroids, and then they assign each object to the cluster with the centroid that is closest according to the Euclidean distance. When all objects have been assigned, the positions of the $k$ centroids are recalculated. The assignment process is repeated until the centroids no longer move.

Elements that can reflect the lithology and are not or are slightly affected by anthropogenic inputs were selected as classification indicators. Complementally, if two or more variables are strongly correlated, it affects the weighting on results. Therefore, variables with a weak or no correlation should be selected in preference. Subsequently, the $k$-means clustering program was performed in MATLAB. During program execution, data are automatically normalized to reduce the dimensional impact. It is known that this algorithm has difficulties in determining the optimal cluster number, and, therefore, a statistical parameter

named silhouette coefficient that can assist in the determination was introduced [29]. The silhouette coefficient of sample *i* is defined as

$$S(i) = \frac{b(i) - a(i)}{\max\{a(i), b(i)\}}$$

(2)

where a(*i*) is the average distance of object *i* to all other objects in the same cluster, and b(*i*) is the minimum distance of sample *i* to other clusters. The average of silhouette coefficients of all samples varies from −1 to 1, and the larger the value is, the better the classification result is. The *k*-means clustering program was performed 5–10 times to ensure a reasonable classification result. The model can be considered valid when the results of the silhouette coefficient and classification map are similar or consistent when obtained twice.

### 2.5. Iterative 3σ-Technique

After sample classification, the iterative 3σ-technique was employed to remove extreme outliers in concentration data of those heavy metals in each cluster, where σ is the standard deviation [30], aiming at calculating the robust mean of the data, which can be regarded as the local background value. If data's skewness and kurtosis are too high, logarithmic transformation is performed first to reduce the effect of data distribution. Then, the iterative 3σ-technique was operated. This technique has two steps: calculating the original data set's mean and standard deviation and then omitting all values beyond the mean ± 3σ interval. This iteration is repeated until no value lies outside this range. Note that the multiple of the standard deviation can be appropriately adjusted according to the actual situation. In other words, a wide interval excludes fewer outliers, while a narrow interval excludes more.

### 2.6. Geoaccumulation Index

To assess and quantify heavy metal pollution in soils, various environmental evaluation indices were established [31,32]. In this study, the geoaccumulation index ($I_{geo}$), an important parameter for identifying the impact of human activities, was employed to evaluate heavy metal pollution in the black soils and can be expressed as

$$I_{geo} = log_2\left(\frac{C_i}{1.5C_b}\right)$$

(3)

where $C_i$ is the measured content of element *i* in a black soil sample, and $C_b$ is the background value of element *i*. The constant 1.5 was introduced to weaken the influence of lithology background on the calculation [33,34]. According to $I_{geo}$ values, seven pollution levels can be defined (Table 2).

**Table 2.** $I_{geo}$ in relation to the classification of pollution levels.

| Indicators | State and Grade Descriptions | | | | | | |
|---|---|---|---|---|---|---|---|
| $I_{geo}$ | $I_{geo} < 0$ | $0 \leq I_{geo} < 1$ | $1 \leq I_{geo} < 2$ | $2 \leq I_{geo} < 3$ | $3 \leq I_{geo} < 4$ | $4 \leq I_{geo} < 5$ | $5 \leq I_{geo}$ |
| Pollution levels | Unpolluted | Unpolluted to moderately polluted | Moderately polluted | Moderately to strongly polluted | Strongly polluted | Strongly to extremely polluted | Extremely polluted |

### 2.7. Positive Matrix Factorization Model

The positive matrix factorization (PMF) model proposed by Paatero and Tapper [35] was employed to identify the source of heavy metals in the black soils, which is a statistical

tool for quantifying the contribution of sources to samples according to the composition or fingerprint of the sources. The PMF model can be expressed as follows:

$$x_{ij} = \sum_{k=1}^{p} g_{ik} f_{ki} + e_{ij} \tag{4}$$

where $x_{ij}$ represents the concentration of species $j$ measured on sample $i$, $p$ is the number of factors, $g_{ik}$ is the relevant contribution of factor $k$ to sample $i$, $f_{kj}$ is the content of species $j$ in factor profile $k$, and $e_{ij}$ is the residual for each sample [36]. Factor contributions and profiles are derived by the PMF model minimizing the objective function $Q$.

$$Q = \sum_{i=1}^{n} \sum_{j=1}^{m} \left( \frac{e_{ij}}{u_{ij}} \right)^2 \tag{5}$$

where $u_{ij}$ is the uncertainty of the species $j$ in sample $i$. $u_{ij}$ is calculated using the following equation:

$$u_{ij} = \begin{cases} \frac{5}{6} \times MDL, \ x_{ij} \leq MDL \\ \sqrt{\left( \sigma_j \times x_{ij} \right)^2 + (0.5 \times MLD)^2}, \ x_{ij} > MDL \end{cases} \tag{6}$$

where $\sigma_j$ is the relative standard deviation of the concentration of species $j$; and MDL represents the method detection limit [37]. In this study, to eliminate the influence of lithology, the PMF model is executed in EPA PMF 5.0 software after the data are normalized by the local background.

## 3. Results and Discussion

### 3.1. Summary Statistics and Spatial Distributions of the Heavy Metals

The average contents of the As, Hg, Cr, Cd, and Pb in the black soils are 10.94, 0.049, 59.40, 0.115, and 26.52 ppm, respectively (Table 3), which are similar to those of the black soils from the urban–rural transition zones of Changchun [6]. The coefficients of variation (Cv) of As, Hg, Cr, Cd, and Pb are 14.20%, 110.61%, 11.55%, 51.58%, and 13.41%, respectively. Among them, Hg and Cd have relatively high Cv values, especially Hg, indicating a significant variation in their contents, which may be caused by human activities or differences in their backgrounds. The skewness and kurtosis were calculated for normal distribution tests of those heavy metals. The former measures the data asymmetry around the sample mean, and the latter measures the degree of tailedness in a distribution. The results show that only Cr follows an approximately normal distribution.

**Table 3.** Statistical parameters for As, Hg, Cr, Cd, and Pb in the black soils.

| Elements | Maximum | Minimum | Average | Median | Std | Cv | Skewness | Kurtosis |
|---|---|---|---|---|---|---|---|---|
| As | 28.40 | 6.10 | 10.94 | 10.80 | 1.55 | 14.20% | 2.68 | 20.01 |
| Hg | 1.434 | 0.009 | 0.049 | 0.038 | 0.05 | 110.61% | 15.17 | 353.65 |
| Cr | 84.60 | 39.90 | 59.40 | 59.30 | 6.86 | 11.55% | 0.07 | −0.25 |
| Cd | 1.760 | 0.038 | 0.115 | 0.106 | 0.06 | 51.58% | 17.81 | 477.41 |
| Pb | 86.59 | 17.58 | 26.52 | 26.16 | 3.56 | 13.41% | 5.67 | 74.46 |

All units are mg/L. Std—standard deviation; Cv—coefficient of variation.

By an ordinary kriging interpolation analysis, the distribution maps of As, Hg, Cr, Cd, and Pb (Figure 2) were obtained. Ranging from cool to warm colors, respectively, the interval values in Figure 2 correspond to the 25th, 50th, and 75th percentiles of the ordered original data of those heavy metals. The areas with high contents of As, Hg, Cd, and Pb are mainly distributed in the southwest of the study area, where the urban areas of Changchun with a high population density and many factories are located. The distribution

characteristics of Hg and Cd are consistent with those reported by Guan et al. [38]. The areas with a high Cr content are mainly located in the east of the study area, which may be caused by the soils there having a high Cr background.

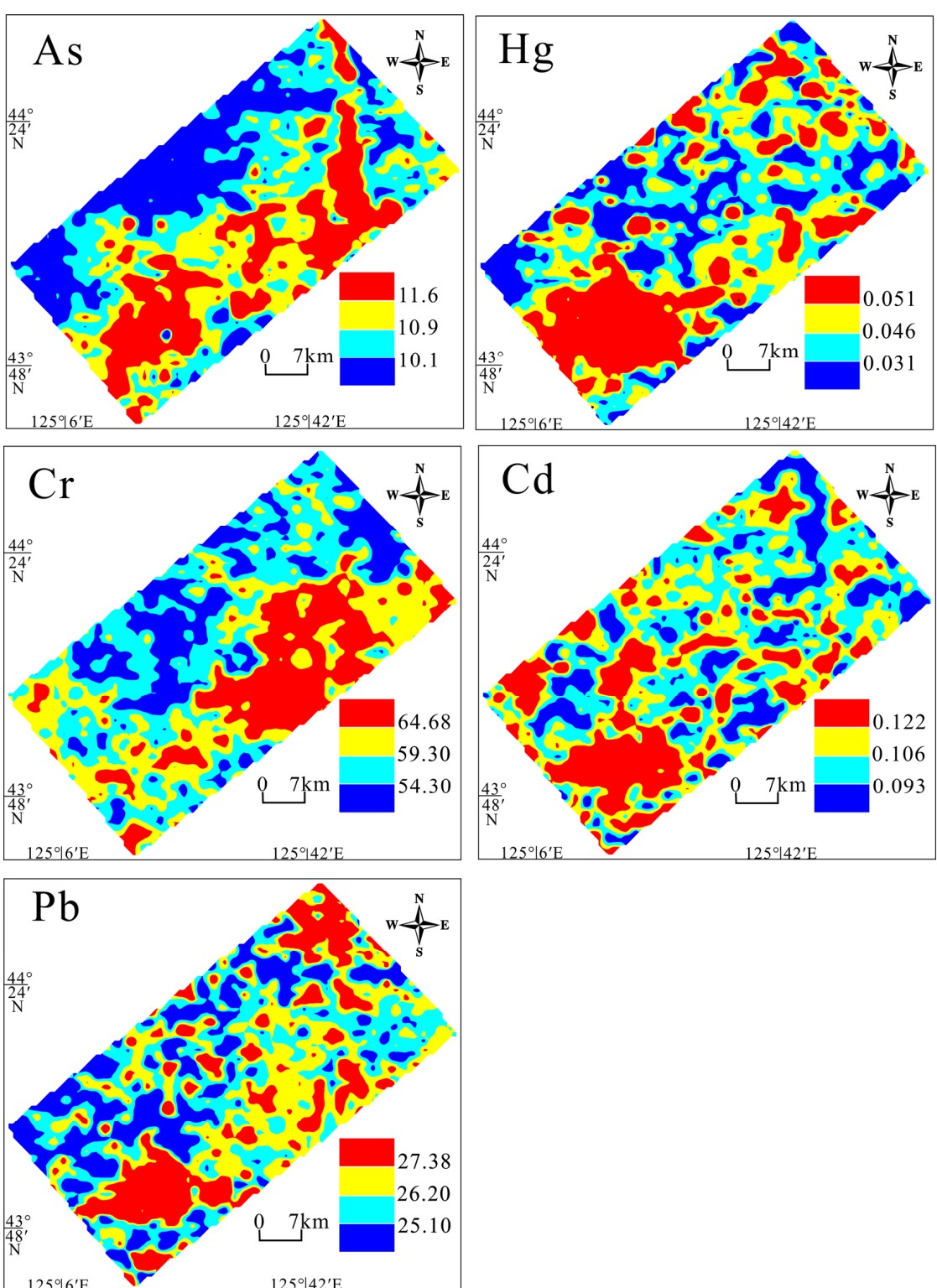

**Figure 2.** The spatial distributions of As, Hg, Cr, Cd, and Pb in the black soils. (All units are mg/L).

### 3.2. Classification Result

Seven elements were selected as classification indicators: $SiO_2$, $Fe_2O_3$, MgO, CaO, Sr, Zr, and Y. In the process of sample classification, a silhouette coefficient graph was obtained (Figure 3). According to the silhouette coefficient rule, the number corresponding to the largest silhouette coefficient should be identified as the optimal cluster number. It can be seen that when the cluster number is 3, the silhouette coefficient reaches the maximum. As a result, the black soil samples were classified into three groups, and, meanwhile, the study area was divided into three geochemical units, each representing a specific lithology (Figure 4). The bedrocks in and around Changchun consist mainly of granite, andesite, basalt, and coarse-grained clastic rocks [23], indicating that the lithology background of the soil parent materials is not very complex, and the classification result is realistic.

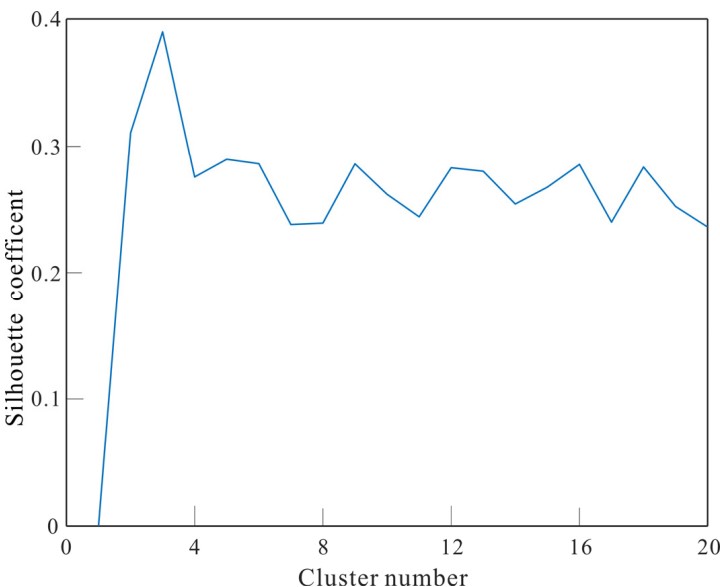

**Figure 3.** The relationship between the silhouette coefficient and the cluster number.

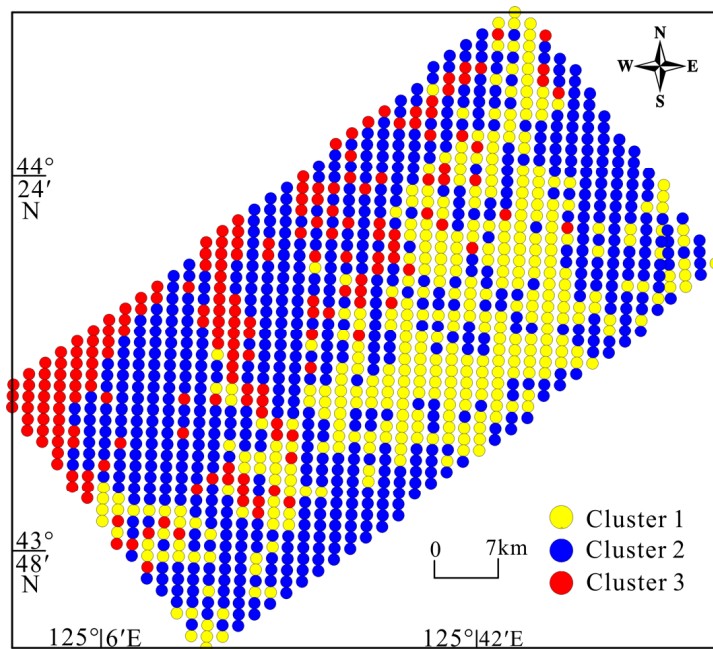

**Figure 4.** The optimal classification map.

There are 361, 692, and 193 samples belonging to clusters 1, 2, and 3, respectively, and the cluster centroids are listed in Table 4. The samples of clusters 1, 2, and 3 are rich in $Fe_2O_3$ and Y; $SiO_2$ and Zr; and CaO, MgO, and Sr, respectively, indicating that their parent materials originated mainly from basic–ultrabasic, acidic, and carbonate rocks, or rocks with similar chemical composition. These three kinds of rocks consist mainly of olivine, pyroxene, and hornblende; quartz, plagioclase, and biotite; and calcite and dolomite, respectively. The mineralogical data of previous studies show that the primary minerals in the study area are mainly plagioclase, quartz, calcite, hornblende, and biotite [23,39], which supports the lithology deduction for each cluster in this paper. The lithological difference can be used to explain the distribution characteristics of those heavy metals in the study area (Figure 2). The location of the high-Cr-content areas corresponds well with that of the cluster 1 samples (Figures 2 and 4), indicating that the enrichment of Cr in the eastern part of the study area may be inherited from the bedrock. However, the high-content areas of chalcophile elements As, Hg, Cd, and Pb, generally found in granite, correspond better with the location of the urban areas in the study area than that of the cluster 2 samples, suggesting that the accumulation of these heavy metals may be anthropogenic in origin. Note that the weathering of carbonate rocks may also be a main reason for the enrichment of Cd in soils [40]. Still, the distribution of the high-Cd-content areas in the study area is quite different from that of the cluster 3 samples (Figures 2 and 4), which also suggests an anthropogenic origin. The cluster 3 samples are mainly located in the western and northern parts of the study area, corresponding well to the location of chernozem (Figures 1 and 4), which has a lime-rich layer of calcium deposits.

**Table 4.** The centroids of the three clusters.

| Elements | Cluster 1 (361) | Cluster 2 (692) | Cluster 3 (193) |
|---|---|---|---|
| $SiO_2$ | 65.18 | 66.75 | 64.58 |
| $Fe_2O_3$ | 5.18 | 4.52 | 4.39 |
| MgO | 1.39 | 1.29 | 1.51 |
| CaO | 1.50 | 1.39 | 3.36 |
| Sr | 175.57 | 179.20 | 213.34 |
| Zr | 284.85 | 327.78 | 310.77 |
| Y | 27.54 | 25.31 | 24.36 |

The unit of $SiO_2$, $Fe_2O_3$, MgO, and CaO is percentage, and that of Sr, Zr, and Y is mg/L. (*n*) is the number of samples.

### 3.3. Background of the Heavy Metals

After sample classification and the elimination of extreme outliers, the local background values of As, Hg, Cr, Cd, and Pb in each cluster were obtained (Table 5). Obviously, each element has three different local background values, which vary between each cluster. For example, the background value of Hg in cluster 2 is 1.3 times as large as that in cluster 3, indicating a significant variation in the background of Hg in the study area. Arsenic, Cr, and Pb have the highest background values in cluster 1, which may be caused by these elements' high abundance in the bedrocks of the cluster 1 samples. It is known that As and Pb can occur as sulfide and Cr as chromite that is concentrated in basic–ultrabasic rocks. The weathering of carbonate rocks may be the main reason why the highest background value of Cd occurs in cluster 3.

For comparison, the uniform background values of those five heavy metals were also estimated by the same method on a regional scale. They were all between the maximum and minimum local background values (Table 5). For example, the uniform background value of Hg is 1.3 times larger than the local background value of cluster 3 but slightly lower than that of cluster 2. Determining heavy metal pollution without considering the lithology influence may lead to some errors. One error is that some high-background areas free from pollution are mistakenly identified as polluted, and another is that some truly polluted low-background areas are ignored. All this prevents soil environmental protection work from reaching a balance between over- and under-protection [42]. Compared with

the risk-screening values (Table 5), both the local and uniform background values of those heavy metals are much lower. However, this does not mean that heavy metal pollution in the black soils is impossible.

**Table 5.** The background values of As, Hg, Cr, Cd, and Pb.

| Items | As | Hg | Cr | Cd | Pb |
|---|---|---|---|---|---|
| Cluster 1 | 11.8 | 0.038 | 64.35 | 0.106 | 26.77 |
| Cluster 2 | 10.6 | 0.039 | 57.06 | 0.108 | 25.97 |
| Cluster 3 | 10.0 | 0.030 | 58.28 | 0.111 | 25.38 |
| Whole region | 10.8 | 0.038 | 59.34 | 0.107 | 26.11 |
| Sedimentary layer * | 5 | 0.11 | 52 | 0.053 | 11 |
| Risk-screening values ** | 30 | 2.4 | 200 | 0.3 | 120 |

All units are mg/L. * The data were from Li [41]. ** The values were from soil environmental quality risk control standard for soil contamination of agricultural land published by Ministry of Ecology and Environment of China in 2018.

### 3.4. Influence of Background Value Selection on the $I_{geo}$ Calculation

As shown by Formula 3, selecting different background values for an element results in different $I_{geo}$ results. The global average of shales and nationwide guideline values, which are generally different from the background values estimated in a particular region, are commonly used in many studies. For example, the background value of the Hg in sedimentary rocks from China is much higher than that estimated in the study area (Table 5). There is no restriction on which background value to use, but one obtained through actual calculation is more appropriate. To show the influence, we took Hg as an example to calculate the $I_{geo}$ by utilizing the background values calculated in the study area and in sedimentary rocks from China (Figure 5).

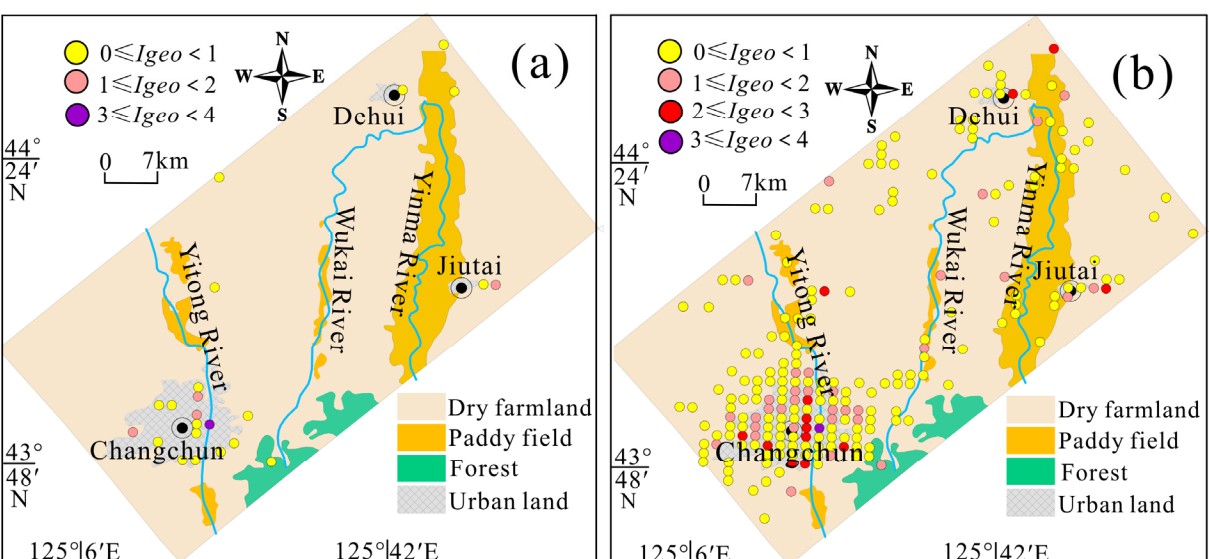

**Figure 5.** The dot plots of Hg $I_{geo}$ calculated from the different background values: (**a**) from the sedimentary rocks in China; (**b**) from the uniform background value in the study area.

The result shows that only 23 pollution spots were found in the study area when the background value of the sedimentary rocks was used, and the pollution level was mainly mild (Figure 5a). However, when changing to the uniform background value, both the amount and degree of Hg pollution increased significantly. Large numbers of pollution spots appeared around the urban areas of Changchun, Dehui, and Jiutai as well as along the Yitong, Wukai, and Yinma rivers (Figure 5b). The comparison results show that the selection of background values has a significant influence on the calculation of the $I_{geo}$ values.

Nevertheless, the lithology influence should be considered when calculating the background values of a particular region, especially in areas with complex lithologies. Then, we compared the $I_{geo}$ values of Hg calculated according to the uniform and local background values, which were estimated before and after classification, respectively. To highlight the difference, we show the pollution sites where the pollution levels have changed (Figure 6). Collectively, the pollution that is easy to misidentify in areas with a high background was eliminated, such as the pollution spots in the A1, A2, and A3 areas with high Hg concentrations (Figures 2 and 6a), and the pollution that is difficult to recognize in areas with a low background was identified, such as those in the B1 and B2 areas with low Hg concentrations (Figures 2 and 6b). The difference would be more significant if the local background values varied over a broader range. As a result, it is better to use the local background values to calculate the $I_{geo}$ of those heavy metals in the study area.

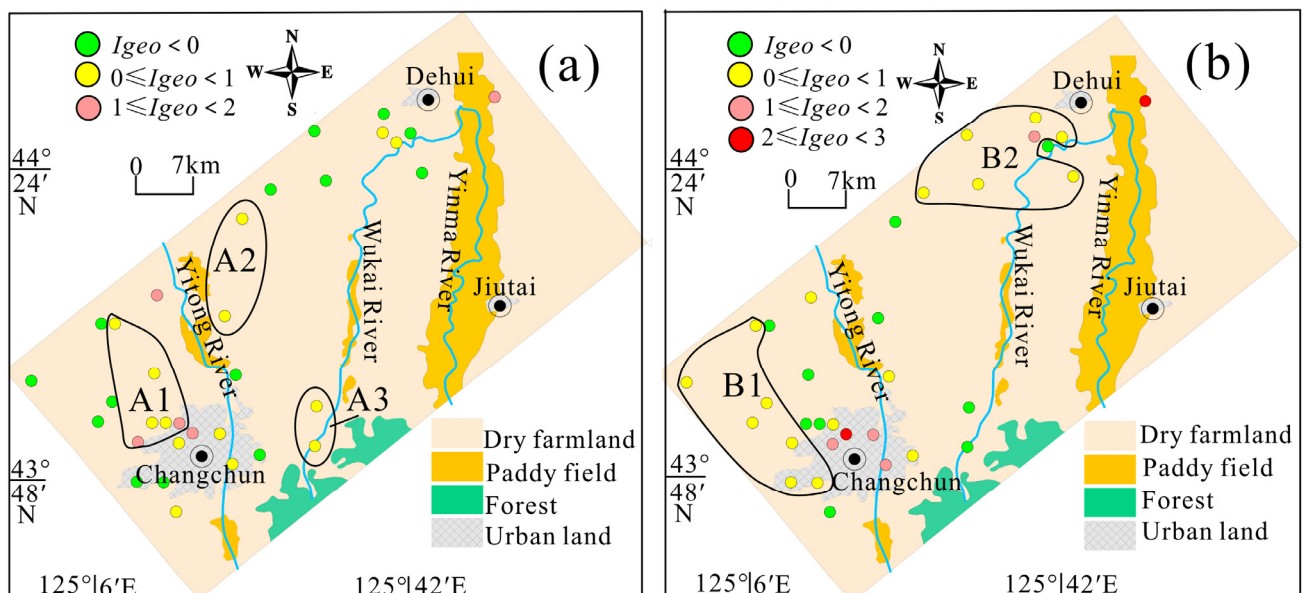

**Figure 6.** The simplified dot plots of Hg $I_{geo}$ calculated from the different background values: (**a**) from the uniform background value; (**b**) from the local background values.

### 3.5. Heavy Metal Pollution in the Black Soils

To evaluate heavy metal pollution in the study area, the $I_{geo}$ values of those heavy metals were calculated according to their local background values. The corresponding dot plots were drawn (Figure 7), except for Cr, as all samples in the study area had Cr-$I_{geo}$ values <0, suggesting that the black soils were free from the pollution of Cr. In terms of As and Pb, only 6 and 12 pollution spots were found in the study area, respectively, and almost all the $I_{geo}$ values were lower than 1, indicating that the pollution degree did not exceed a moderate level (Table 2). This result suggests that anthropogenic activities have little impact on the As, Cr, and Pb concentrations in the black soils.

Lots of black soil samples, however, were contaminated with Hg and Cd, and the number of pollution sites reached 223 and 89, respectively. These two heavy metals have similar pollution distribution characteristics, and the pollution spots are mainly concentrated around cities and along rivers (Figure 7), which is supported by the report of Guan et al. [38]. Collectively, most samples have $I_{geo}$ values of <1, indicating that the pollution degree of most sites is lower than the moderate level (Table 2). Higher levels of contamination, however, occur at some sites in and around the urban areas of Changchun, Jiutai, and Dehui (Figure 7), especially Changchun, and the pollution can reach strong levels at its maximum, which indicates that the accumulation of Hg and Cd was closely associated with human activities.

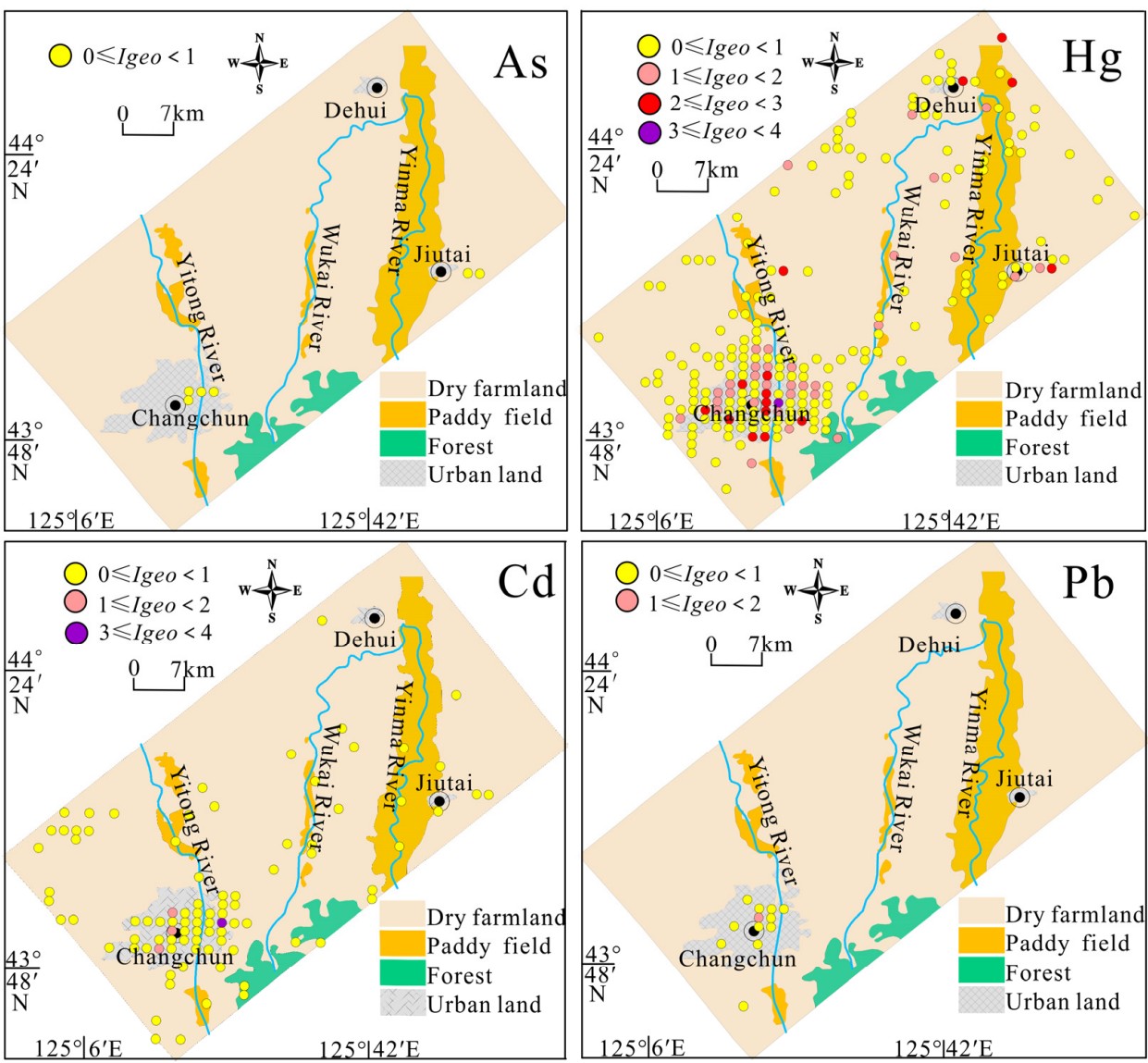

**Figure 7.** The dot plots of $I_{geo}$ for As, Hg, Cd and Pb in the black soils.

The results of $I_{geo}$ show that the black soils in the study area are mainly contaminated with Hg, followed by Cd, and are almost free from the pollution of As, Cr, and Pb. Although the pollution levels of these heavy metals are different, the pollution sites are concentrated in and around the urban areas of Changchun (Figure 7), where factories and automobiles emit mass amounts of waste into the environment, indicating that urbanization may be responsible for the heavy metal pollution in the black soils.

### 3.6. Source Apportionment of the Heavy Metal Pollution

Heavy metals generally have two primary sources in soils: anthropogenic inputs (e.g., agricultural production, mining, and industrial activities) and natural sources (e.g., bedrock and atmospheric sedimentation) [43]. These two sources of heavy metals usually require different treatments. In this study, the positive matrix factorization (PMF) model was employed to identify the sources of the heavy metals in the black soils. Elements N, P, Zr, Y, and Sn, which can reflect specific sources, and As, Hg, Cr, Cd, and Pb were selected for the PMF analysis. Note that the factor number could be adjusted according to the modeling result. Generally, when the values of the scaled residuals for all elements are between −3 and +3, and the difference between $Q_{true}$ and $Q_{robust}$ reaches a minimum, the base run could be considered stable. Finally, four factors that can largely explain the information

in the original data were obtained, and the contribution of the elements is presented in Figure 8. The PMF result indicates that the As, Hg, Cr, Cd, and Pb in the black soils were derived from different sources.

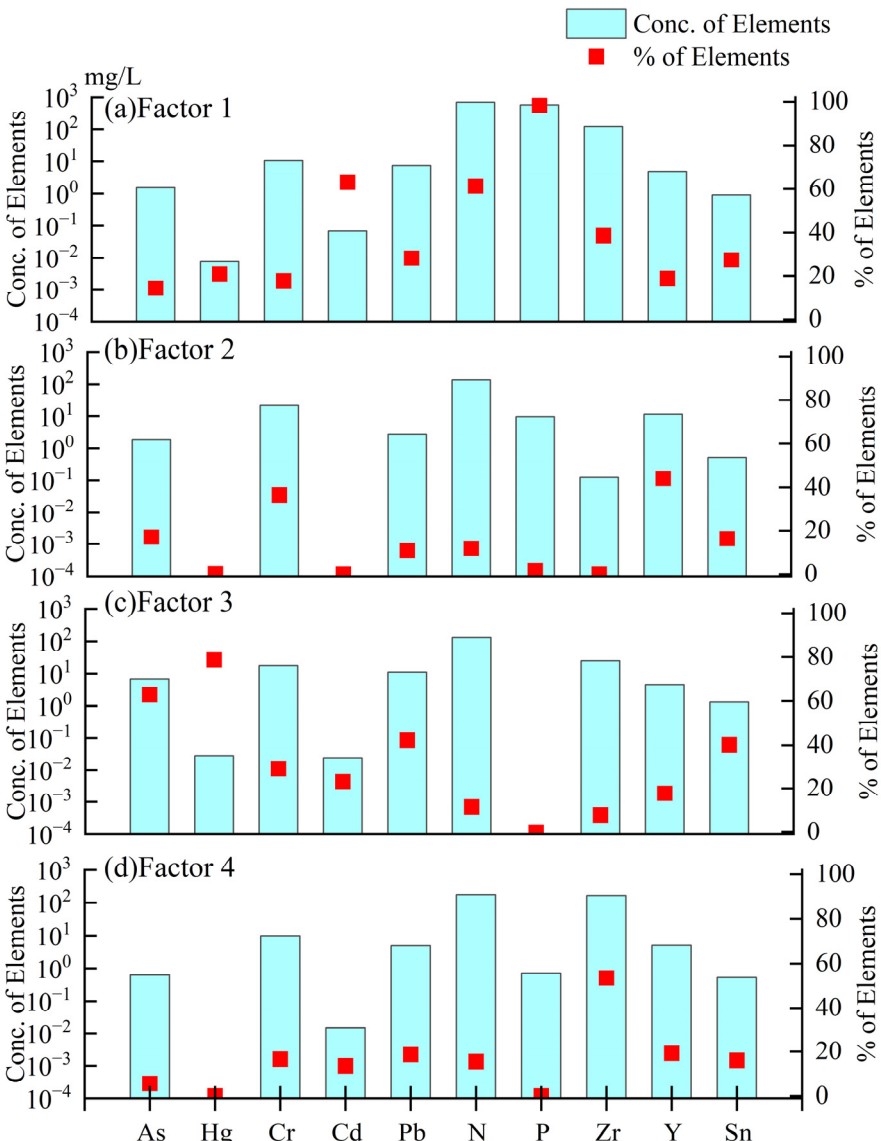

**Figure 8.** Factor profile and concentration percentages from the PMF model: (**a**–**d**) factor 1-4.

Factor 1 has high factor loading values for Cd (63.17%), P (98.28%), and N (61.30%), and N and P are both essential nutrients for crops. The chemical fertilizers used in agricultural production are potential sources of Cd [44]. For example, the raw material of phosphate fertilizer production is phosphate rock, and phosphate rock processing can introduce heavy metal Cd. In addition, some studies report that the use of pesticides on crops can increase the risk of exposure to Cd [31,38], which may be the main reason why there are some Cd pollution spots in the crop regions of the study area. This harmful material may spread to a broader area through groundwater and rainwater, which should be paid attention to. We conclude that agricultural activities are a major source of the Cd pollution in the study area. Incidentally, the contribution of factor 1 to Hg is 20.90%, indicating that agricultural activities may also have contributed to the Hg pollution in the study area. Previous studies found that Hg is significant in pesticides and fertilizers [37]. Thus, factor 1 can be interpreted as an agricultural component.

Factor 2 is dominated by Y and Cr, and their proportions are 44.12% and 36.53%, respectively. Yttrium is a relatively stable rare-earth element, mainly found in the Earth's crust, and is often enriched in the Ti and Fe oxides in soils, which is also supported by the *k*-means clustering result that Y and $Fe_2O_3$ were concentrated in samples of the same cluster. Together, they reflect the lithology of basic–ultrabasic rocks or rocks with a similar chemical composition. Hence, factor 2 can be regarded as a component of this rock, which may be the main source of the Cr in the black soils. This is consistent with the conclusion that parent materials were the main source of the Cr in soils [6,45,46].

Factor 3 is characterized by As (62.96%), Hg (78.73%), Pb (42.25%), and Sn (40.15%). Arsenic and Sn are vital in alloy manufacturing, an essential step in automobile production. Changchun, where the Chinese First Automobile Group Company is located, is an important heavy industry base in northeastern China and has many factories related to automobile production. A field investigation and satellite image observations show that the Hg and As pollution is mainly concentrated in areas with many automobile plants and machinery manufacturing companies, where industrial sewage was reported to cause soil pollution in Changchun [47]. As a result, steel production may be an essential contributor to the Hg and As contamination in the black soils.

Note that, in Changchun, coal is used in large quantities for electricity generation and winter heating, and emissions from burning coal lead to the enrichment of Hg and As through atmospheric deposition [38,45,48]. Obviously, this may also be the main source of the Hg and As in the black soils. Our conclusion is supported by previous studies, which state that the Hg in soils usually comes from nearby metal extraction and processing, coal combustion, and atmospheric deposition [6,49]. In addition, the contribution of factor 3 to Cd is 23.17%, indicating that industrial activities may be a reason for Cd pollution, which may be related to some Cd-electroplating factories in the urban areas of Changchun. Regarding Pb, the pollution spots are mainly distributed in the areas with large traffic flows in Changchun, which the discharge of exhausted gas during traffic activities can explain. In many studies, Pb is used as a major indicator of vehicle emissions [50,51]. As a result, factor 3 was determined to be a component of industrial activities and vehicle emissions.

Factor 4 is weighted primarily on Zr (53.53%), with no loading of other heavy metals. The *k*-means clustering result showed that the Zr in the black soils, like the $SiO_2$, was derived mainly from acid rocks or rocks with a similar chemical composition, and, therefore, factor 4 was considered as a geogenic component.

According to the contribution rate of heavy metals, the four factors can be ranked in descending order: factor 3 (47.23%) > factor 1 (28.90%) > factor 2 (12.91%) > factor 4 (10.96%), which indicates that industrial activities and coal burning contributed the most to the heavy metal pollution in the black soils, followed by agricultural activities. Moreover, the pollution is distributed mainly in and around the urban areas of Changchun. Previous research suggested that urbanization has preferentially increased the content of the heavy metals in soils closer to the urbanized areas [6,50], where the dense populations, factories, and traffic activities are concentrated. We conclude that the pollution of Hg, Cd, Pb, and As in the black soils of the study area is mainly anthropogenic (76.13%), rather than of a geogenic origin (23.87%). This conclusion is consistent with the fact that the study area is widely cultivated with corn and rice and has numerous steel and coal-burning industries.

### 3.7. Implications for Sustainable Development

Black soils are precious natural resources, so keeping them sustainable is important. Sustainable development aims to balance the interests of society, the economy, and the environment; promote sustainable economic growth; protect the ecological environment; and achieve the rational use of resources and sustainable management [52]. Therefore, the principle of sustainable development must be taken into account when solving the problem of the heavy metal pollution in black soils. In the prevention and control of the heavy metal pollution in black soils, effective measures should be taken to reduce the emission of the heavy metal pollution's sources, such as promoting organic agriculture and strengthening

domestic waste treatment. In addition, the natural advantages of black land should be used, as much as possible, to carry out organic planting and ecological farming to reduce the infringement on the environment, improve the productivity and ecological value of black land, and promote the sustainable development of the local economy and society.

## 4. Conclusions

This study employs the *k*-means clustering method to suppress the lithology influence on the calculation of the As, Hg, Cr, Cd, and Pb background in the black soils from the central-eastern part of Changchun. The main findings are as follows:

1. The *k*-means clustering method can effectively eliminate the lithology influence on the calculation of the element background. Each element obtained three different local background values; some were higher than the uniform background value, while some were lower.

2. The choice of different background values significantly impacts the $I_{geo}$ index. According to the application effect, the background priority order is local background value > whole region background value > sedimentary layer background value. In the pollution evaluation, the influence of eliminating lithology should be considered first.

3. The black soils were mainly contaminated with Hg, followed by Cd and Pb, and were almost free from the pollution of Cr and As. Heavy metal pollution is dominantly anthropogenic rather than geogenic in origin. Overall, the current threshold values of these heavy metals are lower than the risk control values of some environmental quality standards, but many polluted sites appear in the black soils, which should be paid attention to.

**Author Contributions:** Conceptualization, Y.S. (Yaoyao Sun) and X.Z.; methodology, X.Z. and L.H.; software, Y.S. (Yaoyao Sun) and X.Z.; validation, Y.Z. and L.H.; investigation, Y.Z. and Q.W.; resources, J.L.; data curation, J.L.; writing—original draft preparation, Y.S. (Yaoyao Sun) and Y.Z.; writing—review and editing, L.H. and X.Z.; supervision, Y.S. (Yanxiang Shi), C.M., and Q.W.; project administration, Y.S. (Yanxiang Shi) and Q.W.; funding acquisition, J.L. All authors have read and agreed to the published version of the manuscript.

**Funding:** This research has been supported by the national key R & D program of China (2021YFC2901801 and 2016YFC0600600).

**Institutional Review Board Statement:** Not applicable.

**Informed Consent Statement:** Not applicable.

**Data Availability Statement:** Data are provided in the article.

**Acknowledgments:** We thank researcher Rongjie Bai with the Geological Survey Institute of Jilin Province for providing research materials. We also would like to thank editor Milanka Hrnjez and four anonymous reviewers for their constructive comments and suggestions, which have improved the paper significantly.

**Conflicts of Interest:** The authors declare that they have no known competing financial interests or personal relationships that could have appeared to influence the work reported in this paper.

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
