# Peer review of "Evaluation and Source Identification of Heavy Metal Pollution in Black Soils, Central-Eastern Changchun, China"

_sustainability, doi:10.3390/su15097419_

Round 1

Reviewer 1 Report

Title: Evaluation and source identification of heavy metal pollution in black soils, central-eastern Changchun, NE China

This paper discusses the heavy metal pollution in black soils collected from the central-eastern part of Changchun, China. Authors have used Geoaccumulation index to investigate the pollution status and interpretations have been made considering the spatial variations, local background levels, lithology influence, etc. The research question has been addressed in a scientific manner adopting an acceptable methodology. The manuscript is well-written and organized. However, the paper needs some improvements before it could be considered for publication.

Some of my general comments are given below:

1)      Include instrument models used for sample analysis.

2)     Include the following information; how the samples were prepared prior to the digestion, the weight of the soil sample, and the fraction used for sample digestion. How the digestions were performed (method, temperature, instruments used, etc.)

3)      I suggest using the standard format of units.

4)      How were the quality assurance and quality control maintained during the analysis? Include relevant information.

5)      What is the software used for spatial mapping? Include relevant details.

Reviewer 2 Report

This manuscript entitled “Evaluation and source identification of heavy metal pollution in black soils, central-eastern Changchun, NE China” deals with the concentrations of As, Hg, Cr, Cd, and Pb in the black soil samples collected from the central-eastern part of Changchun, China, in order to evaluate the heavy metal pollution in the area. A total of 1246 surface soil samples have been collected from the study area. Τhe lithology influence and local background values of As, Hg, Cr, Cd, and Pb in the black soils has been investigated and the pollution status of those potentially toxic elements was explored.

First of all, I suggest that there are other MDPI Journals that are more appropriate for this manuscript (geosciences, land minerals). Besides the conclusion “…many polluted sites appear in the black soils, which should be paid attention to.” there is no link between the content of the manuscript and sustainability.

In analytical methods more details must be presented for the analytical equipment and methods used. For example, what kind of XRF was used? WD or ED. Analytical data for some elements, like alkalis, Ti, Nb, Ni, Co  that can easily by analyzed with the applied analytical methods have not been provided. This data is valuable for the identification of the soil origin end consequently for the evaluation of the local geochemical background of those potentially toxic elements in the study area.

The conclusions of this paper would be strengthened if mineralogical data had been included to support geochemistry in discussion.

Original data should be provided as supplementary material or deposited in an international database and linked to the paper.

Reviewer 3 Report

The study presents the heavy metal contamination assessment of black soils in northern China. The study is essential since the researchers used a machine learning method to identify local background values by eliminating the lithologic influences. However, I recommend the improvement of the manuscript from the points I have listed below.

-        Lines 61-69: Are there any studies conducted to find the local background contaminant values using such statistical/machine learning methods as k-means clustering? If not, it would be better if the authors emphasized this in the text.

-        Lines 140-141: “The k-means clustering program was performed several times to ensure 140 a reasonable classification result.” Several refer to how many, and what are the criteria for “a reasonable classification result”?

-        Lines 297-308: Can there be another reason for high Hg values around the areas highlighted? Are there any possible anthropogenic sources of Hg in the vicinity of hot spots?

-        Figure 7: Hg plot is the one the authors have already given as the Igeo values found using the uniform background values. Can the authors check if there was an error? Or all the plots are drawn using uniform background values?

-        Lines 333-334: When the source apportionment results and the discussion made in section 3.6 is examined, it can be inferred that the sources of not only Hg and Cd but also Pb and As are of anthropogenic origin (Lines 408-409). Then, how come the Igeo calculations yielded the black soils in the study area are “almost free from the pollution of As and Pb”? That’s why I believe using the k-means clustering method to identify local background values tends to introduce a bias in the background values. I understand that the lithologic influences are eliminated using the method. However, the land use in the study region and major heavy metal sources were not considered while applying the method. I wonder if the results would change if the authors used only the samples that can be classified as “background,” i.e., the samples away from any anthropogenic sources and in the opposite direction of the local prevailing wind direction, rather than using all samples from the specific type of black soil.

Reviewer 4 Report

This manuscript was investigated the status and source heavy metal pollution in the black soils, central-eastern Changchun, China. The results concluded that Hg was the main contamination in the black soils, and agricultural activities was the main source of heavy metal pollution. I think the topic is of interest and better understanding of the effects of heavy metal pollution on black soils.

Several amendments are suggested as follows:

Line 3, the “NE” can be deleted in the Title.

Line 20, this sentence should be revised.

Lines 171-178, this paragraph is not suitable to be placed here.

Please check the units of the element content (mg/L) in Table 3-5.

Figure 8, “Species” may be changed to “Elements”. And the units of the elements need to provide.

I do not know whether the factors interpreted as agricultural, industrial activities, and coal burning is rational. The pollution distribute mainly around and in the urban area of Changchun, so is it right that agricultural activities is the main source?

The discussion should be improved, especially by providing evidence support.

Round 2

Reviewer 2 Report

As I've seen only the Materials and Method section has been improved...
